# Reference-Based POMDPs

**Edward Kim**
School of Computing
Australian National University
Canberra, Australia
edward.kim@anu.edu.au

**Yohan Karunanayake**
School of Computing
Australian National University
Canberra, Australia
yohan.karunanayake@anu.edu.au

**Hanna Kurniawati**
School of Computing
Australian National University
Canberra, Australia
hanna.kurniawati@anu.edu.au

## Abstract

Making good decisions in partially observable and non-deterministic scenarios is a crucial capability for robots. A Partially Observable Markov Decision Process (POMDP) is a general framework for the above problem. Despite advances in POMDP solving, problems with long planning horizons and evolving environments remain difficult to solve even by the best approximate solvers today. To alleviate this difficulty, we propose a slightly modified POMDP problem, called a Reference-Based POMDP, where the objective is to balance between maximizing the expected total reward and being close to a given reference (stochastic) policy. The optimal policy of a Reference-Based POMDP can be computed via iterative expectations using the given reference policy, thereby avoiding exhaustive enumeration of actions at each belief node of the search tree. We demonstrate theoretically that the standard POMDP under stochastic policies is related to the Reference-Based POMDP. To demonstrate the feasibility of exploiting the formulation, we present a basic algorithm REFSOLVER. Results from experiments on long-horizon navigation problems indicate that this basic algorithm substantially outperforms POMCP.

## 1 Introduction

Computing motion strategies that are robust to uncertainty is a fundamental problem in robotics. A robot often needs to strategically plan in an environment where it only has partial knowledge about itself and the state of the world due to actuator disturbances, limited knowledge about the environment, and imperfect sensory information. Such planning problems can generally be framed as a Partially Observable Markov Decision Process (POMDP) [7, 22], which is a general mathematical framework for planning in partially observable and non-deterministic scenarios. POMDPs explicitly model partial observability using a probability distribution over the state space, called a *belief*, and compute the best strategy with respect to beliefs. Finding the exact solution to a POMDP problem is computationally intractable in the worst case [15, 17] for two main reasons. Firstly, the size of the belief space grows exponentially with respect to the size of the state space. Secondly, the number of action-observation histories grows exponentially with respect to the planning horizon. These challenges are known as the *curse of dimensionality* and the *curse of history* respectively. To deal with these challenges, many practical solvers have been proposed in the past few decades [9]. Sampling-based approximate solvers [12, 21, 29] have made good progress on both curses by employing particle filters to maintain a belief search tree and judiciously exploring a *subset* of histories in this search tree.

37th Conference on Neural Information Processing Systems (NeurIPS 2023).

Notwithstanding these advances, some partially observed problems, such as those involving long planning horizons and progressively evolving environments remain relatively difficult to solve by even the best approximate POMDP solvers today. In particular, all of the aforementioned solvers struggle with a planning horizon of more than 15 steps primarily because, to infer the best action, they rely on exhaustive enumeration of actions at each belief node in the search tree.

To alleviate the above difficulties, and hence further improve POMDP solving capabilities, we propose the idea of a *Reference-Based POMDP*. A Reference-Based POMDP is a modification of the standard POMDP where, in addition to maximising total rewards, an agent's objective function is penalised for deviating too far from a given stochastic reference policy. This approach leads to an interesting interpretation where the agent must trade off between two possibly competing objectives. The agent should respect the given initial stochastic policy unless deviating substantially from it leads to much higher rewards.

A key outcome of this formulation is that solving a Reference-Based POMDP does not require exhaustive enumeration of the actions at each belief node. Rather, optimisation can be performed analytically because: (i) the space of policies is relaxed to include *stochastic policies*; and (ii) the form of the penalty is represented as the KL divergence from the reference policy. Under suitable transformations, this leads to a linear Bellman backup (Theorem 3.1) where the sampling distribution is the reference policy itself. As the reference policy is known a priori, the above insight facilitates algorithms where the actions can be sampled (as opposed to enumerated) at each step which, in turn, can be efficiently computed using Monte Carlo methods.

This work can be viewed as an extension of a body of literature related to Linearly Solvable Optimal Control[24, 26, 27, 1, 19, 28, 3] in fully observable environments (see Section 7 for a brief summary) to partially observable ones. While it is true that a POMDP can be viewed as an MDP in the belief space [7], extending the aforementioned works to partially observable domains is non-trivial because of the infinite and continuous nature of the belief space and the difficulty of computing belief updates. To the authors' knowledge, the approaches outlined above have not yet been extended to partially observable domains.

The paper is organised as follows. Section 2 recounts the basic setup of a POMDP. Section 3 introduces the Reference-Based POMDP and outlines the relevant theory. Section 4 demonstrates that the standard POMDP can be related to the Reference-Based POMDP via an embedding. Section 5 presents REFSOLVER a preliminary approximate solver for the Reference-Based POMDP. Experimental results on long-horizon 2D and 3D navigation problems are presented in Section 6 and indicate that our solver can be employed to substantially outperform POMCP [21]. Note that the problems we have presented, are not trivial due to the long planning horizon and the challenges of partial observability and transition uncertainty. Finally, we outline the relevant literature in Section 7 and summarise the contributions, limitations and future directions of the paper in Section 8.

## 2   POMDP Preliminaries

In this paper, we focus on an infinite-horizon formulation of the POMDP. Formally, a standard POMDP is completely specified by a tuple $\langle \mathcal{S}, \mathcal{A}, \mathcal{O}, \mathcal{Z}, \mathcal{T}, R, \gamma \rangle$. Here, $\mathcal{S}$ denotes the set of all possible states of the agent and its environment, $\mathcal{A}$ denotes the set of all possible actions the agent can perform, and $\mathcal{O}$ denotes all the possible observations that the agent can perceive. Although our transformation could be applied to POMDPs with continuous state, action, and observation spaces, for simplicity, in this paper, we will assume that $\mathcal{S}$, $\mathcal{A}$ and $\mathcal{O}$ are all discrete.

At each time step, the agent occupies a state $s \in \mathcal{S}$ and executes an action $a \in \mathcal{A}$, after which its state transitions to a new state $s' \in \mathcal{S}$. The outcome of this execution is non-deterministic, and this uncertainty is represented in the Markovian transition function $\mathcal{T}$ as a conditional probability function $\mathcal{T}(s' \,|\, s, a)$. At the new state $s'$, the agent perceives an observation $o \in \mathcal{O}$. Uncertainty in the observation perceived – having undertaken action $a \in \mathcal{A}$ and transitioned to state $s'$ – is described by the conditional probability function $\mathcal{Z}(o \,|\, s', a)$, referred to as the observation function. The agent's action incurs a reward, which is defined by a bounded real-valued function $R : \mathcal{S} \times \mathcal{A} \to \mathbb{R}$. The parameter $\gamma \in (0, 1)$ represents the discount factor.

As the agent does not know its true state, at each time step, the agent maintains a *belief* about its state. A belief, denoted as $b$, is a probability distribution over the state space $\mathcal{S}$. The set of all possible

beliefs forms a belief space of $(|\mathcal{S}| - 1)$-dimensional simplex, which we denote as $\mathcal{B}$. The belief the agent maintains is updated via Bayesian inference using the transition and observation models $\mathcal{Z}$ and $\mathcal{T}$, respectively. The exact belief update is completely determined by an action-observation pair. Formally, if $b' = \tau(b, a, o)$ denotes the agent's next belief after taking action $a$ and perceiving observation $o$, then $b'$ is defined as:

$$b'(s') = \eta \, \mathcal{Z}(o \,|\, s', a) \sum_{s \in \mathcal{S}} \mathcal{T}(s' \,|\, s, a) \, b(s) \tag{1}$$

where $\eta$ is a normalising factor. The solution to a POMDP problem is an optimal policy $\pi^* : \mathcal{B} \to \mathcal{A}$ that maximises the value function, defined as:

$$V^*(b) = \max_{a \in \mathcal{A}} \left[ R(b, a) + \gamma \sum_{o \in \mathcal{O}} P(o \,|\, a, b) V^* \big( \tau(b, a, o) \big) \right] \tag{2}$$

where

$$R(b, a) := \sum_{s} R(s, a) \, b(s) \tag{3}$$

is the expected reward under the belief $b$. The notation $P(o \,|\, a, b)$ is the probability the agent perceives an observation $o \in \mathcal{O}$ having performed the action $a \in \mathcal{A}$ under the belief $b$, and is defined as

$$P(o \,|\, a, b) := \sum_{s' \in \mathcal{S}} \mathcal{Z}(o \,|\, s', a) \Big( \sum_{s \in \mathcal{S}} \mathcal{T}(s' \,|\, s, a) b(s) \Big). \tag{4}$$

Computing the exact solution $\pi^*$ and optimal value function $V^*$ is intractable [14, 15, 17] in general, and therefore state-of-the-art solvers approximate solutions via sampling [9].

## 3 Reference-Based POMDP

We now extend the analytical approach for the Linearly Solvable MDP (see Appendix Section A.1) to the POMDP. To this end, we define a *Reference-Based POMDP*, which is specified by the tuple $\langle \mathcal{S}, \mathcal{A}, \mathcal{O}, \mathcal{Z}, \mathcal{T}, R, \gamma, \bar{U} \rangle$. Similar to the standard POMDP, $\mathcal{S}$, $\mathcal{A}$, and $\mathcal{O}$ refer to the state, action, and observation spaces respectively. We denote by $\mathcal{Z}$, $\mathcal{T}$, and $R$ the observation, transition, and reward functions, respectively, while $\gamma \in (0, 1)$ denotes the discount factor.

**Assumption 3.1.** *A* reference belief-to-belief transition probability*:*

$$\bar{U}(\cdot, \cdot \,|\, b) \in \Delta(\mathcal{A} \times \mathcal{O}) \tag{5}$$

*is given for all $b \in \mathcal{B}$.*

This reference transition is assumed by the problem and could be constructed in multiple ways. At the most abstract the level this could be given in a top-down fashion (e.g. an offline policy that has been generated for a similar version of a POMDP problem). On the other hand, it could also be constructed from the ground up. For example, one could take a fully observed policy $\pi^{\text{FO}} : \mathcal{S} \to \mathcal{A}$ and infer a reference stochastic action $\bar{\pi}(\cdot \,|\, b)$ according to

$$\bar{\pi}(a \,|\, b) := \eta \Big[ \alpha \sum_{s \in \mathcal{S}} I_{\{\pi^{\text{FO}}(s) = a\}} b(s) + (1 - \alpha) \, \rho(a \,|\, b) \Big] \tag{6}$$

where $\rho$ is some default distribution for sampling actions and the parameter $\alpha \in (0, 1)$ indicates the saturation level of the fully observed policy. The exact choice of fully observed policy depends on the context. Examples might include: an MDP policy for the fully observed version of the problem, a policy induced by feasible paths generated by motion planners, a policy induced by all source shortest path in the case of a navigation problem, and even a policy representing the passive dynamics of a system at any given state. Regardless of how the reference stochastic action is constructed, in a ground-up construction, the reference belief-to-belief transition probability will take on the more specific form:

$$\bar{U}(a, o \,|\, b) := P(o \,|\, a, b) \, \bar{\pi}(a \,|\, b) \tag{7}$$

where $P(o \,|\, a, b)$ is given by (4).

Inspired by Linearly-Solvable MDPs (Section A.1), a Reference-Based POMDP asserts that, for any belief $b \in \mathcal{B}$, we can realise "belief-to-belief" transition probabilities $U(\cdot, \cdot \,|\, b) \in \Delta(\mathcal{A} \times \mathcal{O})$ of

our choosing. Note that this is an *ideal* transition probability which may not actually be realisable in practice even by a stochastic policy (see Section 3.3 and Section 8 for further commentary). To prohibit arbitrary transitions, at any belief $b \in \mathcal{B}$, we will further restrict our choice of distribution $U$ to one that satisfies the property

$$\bar{U}(a, o \,|\, b) = 0 \implies U(a, o \,|\, b) = 0 \quad \forall a \in \mathcal{A}, o \in \mathcal{O}.$$

Denote the class of such distributions by $\mathscr{U}(b) \subset \Delta(\mathcal{A} \times \mathcal{O})$ which will form our admissible reference transition probabilities (which can be viewed as admissible controls) from a given belief. Then, $\mathscr{U} := \bigcup_{b \in \mathcal{B}} \mathscr{U}(b)$ is the class of all admissible reference transition probabilities. Similar to (25), we penalise choices of $U(\cdot, \cdot \,|\, b)$ that represent large deviations from $\bar{U}(\cdot, \cdot \,|\, b)$ using KL divergence: $\mathrm{KL}\left(U(\cdot, \cdot \,|\, b) \,\|\, \bar{U}(\cdot, \cdot \,|\, b)\right)$.

Now, the above reference transition probability affects the definition of the reward function too. At the most abstract level, the reward function $R : \mathcal{B} \times \mathcal{A} \to \mathbb{R}$ is parameterised by belief–action pairs. Hence, the expected immediate cost of taking a stochastic belief-to-belief transition governed by $U(\cdot, \cdot \,|\, b) \in \mathscr{U}(b)$ is given by

$$R(b, U) := \sum_{a,o} R(b, a) U(a, o \,|\, b) \tag{8}$$

where again $R(b, a) := \sum_{s \in \mathcal{S}} R(s, a) b(s)$. The solution to a Reference-Based POMDP problem is then an optimal stochastic policy $\pi^* : \mathcal{B} \to \mathscr{U}$ that maximises:

$$\mathcal{V}^*(b) = \sup_{U(\cdot, \cdot \,|\, b) \in \mathscr{U}(b)} \Big( R(b, U) - \mathrm{KL}\left(U(\cdot, \cdot \,|\, b) \,\|\, \bar{U}(\cdot, \cdot \,|\, b)\right)$$
$$+ \gamma \sum_{a,o} U(a, o \,|\, b) \mathcal{V}^*\big(\tau(b, a, o)\big) \Big). \tag{9}$$

## 3.1 Existence and Uniqueness of the Solution

One might ask if the reformulation (9) is satisfied uniquely by some function $\mathcal{V} : \mathcal{B} \to \mathbb{R}$. The following lemma ensures that there exists a unique (up to $\| \cdot \|_\infty$-equivalence) bounded function $\mathcal{V} : \mathcal{B} \to \mathbb{R}$ satisfying the Bellman equation. Its proof is deferred to Appendix A.2.

**Lemma 3.1.** *Consider the Banach space $\mathbb{B}(\mathcal{B})$ of bounded functions $\mathcal{V} : \mathcal{B} \to \mathbb{R}$ equipped with the supremum norm $\| \cdot \|_\infty$. Let $\Phi : \mathbb{B}(\mathcal{B}) \to \mathbb{B}(\mathcal{B})$ be a self-mapping defined by the Reference-Based POMDP backup operator described by equation (9). Then, for $\gamma \in (0, 1)$, the mapping $\Phi$ is a contraction and*

$$\|\Phi \mathcal{V}_1 - \Phi \mathcal{V}_2\|_\infty \le \gamma \|\mathcal{V}_1 - \mathcal{V}_2\|_\infty. \tag{10}$$

Hence, by the Banach fixed point theorem [8], the solution to the Bellman equation exists uniquely in this space whenever $\gamma \in (0, 1)$.

## 3.2 Transforming the Value Function

Here, we show that under an appropriate transformation, the value function of a Reference-Based POMDP (9) can also be linearised. Its proof is given in Appendix A.3.

**Theorem 3.1.** *Define the transformation $\mathcal{W}(b) := e^{\mathcal{V}(b)}$ for any $b \in \mathcal{B}$. The value function (9) of the Reference-Based POMDP $\langle \mathcal{S}, \mathcal{A}, \mathcal{O}, \mathcal{Z}, \mathcal{T}, R, \gamma, \bar{U} \rangle$ is equivalent to*

$$\mathcal{W}(b) = \sum_{a,o} \bar{U}(a, o \,|\, b) e^{R(b,a)} \mathcal{W}^\gamma\big(\tau(b, a, o)\big). \tag{11}$$

*The solution $\mathcal{W}^*$ to equation (11) exists and is unique. And, the optimal stochastic "belief-to-belief" transition to the Bellman equation (9) is given by*

$$U^*(a, o \,|\, b) = \frac{\bar{U}(a, o \,|\, b) e^{R(b,a)} \mathcal{W}^{*\gamma}\big(\tau(b, a, o)\big)}{\mathcal{D}[\mathcal{W}^{*\gamma}](b)} \tag{12}$$

*where $\mathcal{D}[\mathcal{W}^{*\gamma}](b) := \sum_{\hat{a}, \hat{o}} \bar{U}(\hat{a}, \hat{o} \,|\, b) e^{R(b,\hat{a})} \mathcal{W}^{*\gamma}\big(\tau(b, \hat{a}, \hat{o})\big)$ is a normaliser.*

Intuitively, the solution $\mathcal{W}^*$ represents the desirability of a given belief (or history of action-observation sequences) where, (12) represents distorting the reference dynamics to higher desirability scores. Moreover, Theorem 3.1 shows that we can converge to $\mathcal{W}^*$ by iterating the transformed Reference-Based POMDP backup operator (11). Lemma 3.1 indicates that the speed of convergence depends on the size of $\gamma$ with faster convergence as $\gamma$ tends to 0. Moreover, Theorem 3.1 also implies that once $\mathcal{W}^*$ is computed, $U^*$ can be computed *exactly* via (12).

Unlike the linearly solvable MDP, exactly computing $\mathcal{W}^*$ is still intractable in practice due to the continuity of the belief space and the cost of belief estimation. However, if we leverage the advances made by sampling-based POMDP solvers and assume that the belief space is sufficiently represented by a finite number of beliefs, then this quantity can be approximated. Of course, finding such a set of beliefs is in itself difficult. Nevertheless, the fact that the optimal value function of a Reference-Based POMDP is equivalent to (11) implies that the optimal value function of Reference-Based POMDP can be computed by purely computing expectations under the reference transition probabilities recursively, which can be approximated efficiently (e.g. via Monte Carlo approximation).

### 3.3 Recovering the Stochastic Actions

While restricting the agent's transition probabilities to $\mathscr{U}(b)$ limits the agent's level of control, $U^*$ as given in (12) is still an *ideal* transition probability and may therefore not be realisable under a stochastic action in general. In other words, one cannot guarantee the existence of a stochastic action $\pi(a \mid b)$ such that $U(a, o \mid b) := P(o \mid a, b)\, \pi(a \mid b) = U^*(a, o \mid b)$ for every $a \in \mathcal{A}$ and $o \in \mathcal{O}$.

Nevertheless, we can still find a $\pi(a \mid b)$ such that $U$ and $U^*$ are close in some sense. A suitable way to formulate the problem is to minimise the KL divergence between $U^*$ and $U$. That is, for each $b \in \mathcal{B}$, we minimise

$$\sum_{a,o} P(o \mid a, b)\, \pi(a \mid b) \log \left[ \frac{P(o \mid a, b)\, \pi(a \mid b)}{U^*(a, o \mid b)} \right] \tag{13}$$

over all stochastic actions $\pi(\cdot \mid b) \in \Delta(\mathcal{A})$. Solving the constrained minimisation problem using Lagrange multipliers yields the following result. See Appendix A.4 for its proof.

**Proposition 3.1.** *The minimiser of* (13) *is*

$$\pi^*(a \mid b) = \frac{\exp[-\Pi(a \mid b)]}{\sum_{\hat{a} \in \mathcal{A}} \exp[-\Pi(\hat{a} \mid b)]} \quad \forall a \in \mathcal{A} \tag{14}$$

*where*

$$\Pi(a \mid b) := \sum_o P(o \mid a, b) \log \left[ \frac{P(o \mid a, b)}{U^*(a, o \mid b)} \right]. \tag{15}$$

## 4 Relating the Reference-Based POMDP and the Standard POMDP

In A.1, we recount a procedure for embedding the standard MDP $\langle \mathcal{S}, \mathcal{A}, \mathcal{T}, R, \gamma \rangle$ inside the reformulated MDP $\langle \mathcal{S}, p, \rho, \gamma \rangle$. The key advantage of this is to apply the more efficient machinery of the reformulation to quickly determine good approximate solutions to the standard MDP. Furthermore, it establishes a theoretical basis to relate the two formulations. Here, we present an analogous embedding for the POMDP. Proofs are deferred to Section A for conciseness.

Suppose an initial belief $b_0 \in \mathcal{B}$ is given and let $\mathscr{R}_{b_0}$ be the set of all reachable beliefs from $b_0$. Let $\mathbb{B}(\mathscr{R}_{b_0})$ denote the set of all bounded functions from $\mathscr{R}_{b_0}$ to $\mathbb{R}$. Consider the family of Reference-Based POMDPs with purely belief-dependent rewards $\rho \in \mathbb{B}(\mathscr{R}_{b_0})$.

$$\{\langle \mathcal{S}, \mathcal{A}, \mathcal{O}, \mathcal{Z}, \mathcal{T}, \rho, \gamma, \bar{U} \rangle\}_{\rho \in \mathbb{B}(\mathscr{R}_{b_0}), \bar{U} \in \mathscr{U}} \tag{16}$$

and a standard POMDP of the form

$$\langle \mathcal{S}, \mathcal{A}, \mathcal{O}, \mathcal{Z}, \mathcal{T}, R, \gamma \rangle. \tag{17}$$

For fixed $(\rho, \bar{U}) \in \mathbb{B}(\mathscr{R}_{b_0}) \times \mathscr{U}$, their respective Bellman equations are:

$$\mathcal{V}(b) = \sup_{U(\cdot, \cdot \mid b) \in \mathscr{U}(b)} \Big( \rho(b) - \mathrm{KL}\left( U(\cdot, \cdot \mid b) \,\|\, \bar{U}(\cdot, \cdot \mid b) \right)$$

$$+ \gamma \sum_{a,o} U(a, o \mid b) \mathcal{V}\big( \tau(b, a, o) \big) \Big) \tag{18}$$

and

$$V(b) = \max_{a \in \mathcal{A}} \Big[ R(b, a) + \gamma \sum_{o \in \mathcal{O}} P(o \,|\, a, b) V\big(\tau(b, a, o)\big) \Big] \tag{19}$$

for all $b \in \mathscr{R}_{b_0}$.[1]

**Definition 4.1.** *Let $\mathbb{B}(\mathscr{R}_{b_0})$ be the set of all bounded functions from $\mathscr{R}_{b_0}$ to $\mathbb{R}$. We say that the pair $(\rho, \bar{U}) \in \mathbb{B}(\mathscr{R}_{b_0}) \times \mathscr{U}$ is an* embedding *of the standard POMDP* (17) *in the Reference-Based POMDP* (16) *with purely belief-dependent rewards if, for each $\hat{a} \in \mathcal{A}$ and $b \in \mathcal{B}$ there exists a $U^{\hat{a}}(\cdot, \cdot \,|\, b) \in \mathscr{U}$ such that:*

$$\sum_{a \in \mathcal{A}} U^{\hat{a}}(a, o \,|\, b) = P(o \,|\, \hat{a}, b) \tag{20}$$

*and*

$$R(b, \hat{a}) = \rho(b) - \mathrm{KL}(U^{\hat{a}}(\cdot, \cdot \,|\, b) \,\|\, \bar{U}(\cdot, \cdot \,|\, b)). \tag{21}$$

**Proposition 4.1.** *Suppose $\rho \in \mathbb{B}(\mathscr{R}_{b_0})$ is such that*

$$\rho(b) = \log \Big[ \sum_{\hat{a} \in \mathcal{A}} e^{R(b, \hat{a})} \Big] \tag{22}$$

*holds where $R(b, \hat{a}) := \sum_{s \in \mathcal{S}} R(s, \hat{a}) b(s)$. Then, if*

$$\bar{U}(a, o \,|\, b) := P(o \,|\, a, b) \frac{e^{R(b, a)}}{\sum_{\hat{a} \in \mathcal{A}} e^{R(b, \hat{a})}} \quad \forall b \in \mathcal{B} \tag{23}$$

*the pair $(\rho, \bar{U}) \in \mathbb{B}(\mathscr{R}_{b_0}) \times \mathscr{U}$ is an embedding of the standard POMDP in the Reference-Based POMDP with purely belief-dependent rewards.*

## 5 An Online Planning Algorithm

We have developed a preliminary on-line solver REFSOLVER for a Reference-Based POMDP to demonstrate some of the advantages of the formulation presented in Section 3. Pseudo code for an implemented algorithm where the reference policy is constructed using a fully-observed policy $\pi^{\mathrm{FO}}$ is presented in the Appendix Section A.6 Algorithm 1. Note that although our description of REFSOLVER is very specific in terms of the reference policy and reference stochastic action, they can readily be replaced with other types of policies and induced reference stochastic actions.

Similar to many state-of-the-art on-line POMDP solvers (e.g., [21, 29]), REFSOLVER constructs a search tree of action-observation histories and uses particles to represent the beliefs. The key insight of REFSOLVER, however, is to exploit the additional information afforded by the fully observed policy to limit exploring every possible action at each history node. Instead, as each backup of the transformed value (9) is essentially an expectation under the reference policy, actions are sampled using the reference policy and the backup is estimated via a straightforward Monte Carlo method. This allows sparser sampling and simulations to resemble a depth-first search traversing to a pre-specified depth before rolling out the leaf nodes using the fully observed policy. Hence, rather than expanding all the actions as POMCP [21], ABT[12] and DESPOT [29] do, REFSOLVER constructs a deeper tree thereby exploiting more information about the long-term behaviour of the system.

To allow for REFSOLVER to also search forward for actions that are not supported by the fully observed policy, some additional noise $\rho$ can be added according to (6) so that $(1 - \alpha)$ can be interpreted as an exploration constant. In our implementation the noise $\rho$ is a uniform distribution over the actions. The parameter $(1 - \alpha)$ can thus be interpreted as REFSOLVER's belief that the fully observed policy can be trusted.

## 6 Experiments

We tested REFSOLVER against the implementation of POMCP [21] in the `pomdp_py` library [4] in two different $60 \times 60$ long-horizon grid-world navigation problems. As REFSOLVER is written in Python, the implementation of POMCP was de-Cythonised for a fair comparison.

---

[1]Note that $\mathscr{R}_{b_0}$ is common to both formulations as their observation and transition models are the same.

We did not compare with DESPOT [29] because the problem scenarios require long horizons and the planning horizon that DESPOT can perform is much lower than POMCP because of its requirement to construct a tree of size $|\mathcal{A}|^d K^d$ where $d$ is the depth of the DESPOT (this difficulty has also been observed in [6]). Given the long horizon nature of the problems, one may question comparison with solvers specifically designed for long horizon problems (as summarised in [9]). Most, if not all, of such solvers rely on reducing the effective planning horizon by constructing macro actions or sub-policies. Since REFSOLVER does not change the overall structure of action and observation spaces and transition and observation functions, it is trivial to apply most techniques that reduce the effective planning horizon to Reference-Based POMDP solvers (including REFSOLVER), which in turn will further improve the performance of REFSOLVER too.

All experiments were performed on a desktop computer with an 8 Core Intel Xeon Silver 4110 Processor and 128GB DDR4 RAM.

## 6.1   Problem Scenarios and Testing Parameters

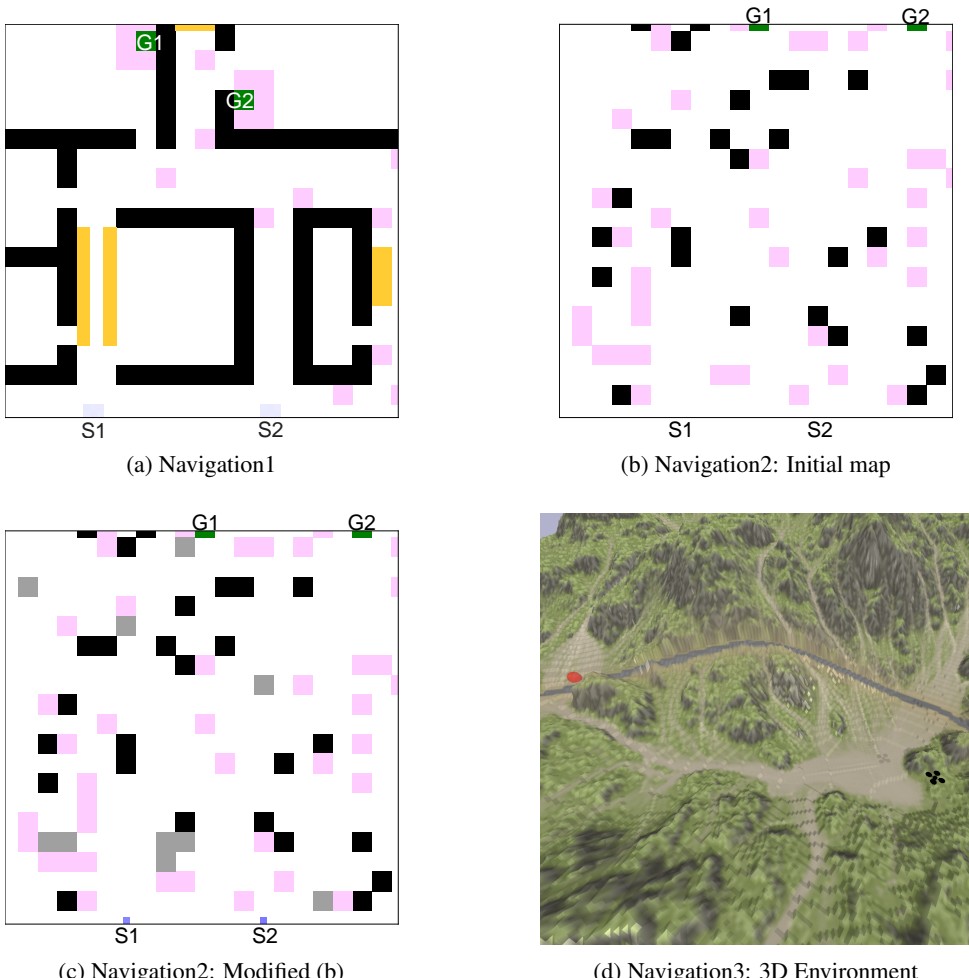

(a) Navigation1

(b) Navigation2: Initial map

(c) Navigation2: Modified (b)

(d) Navigation3: 3D Environment

Figure 1: Illustration of testing scenarios. (a) Navigation1: landmarks (pink), goal states (green), obstacles (black), danger zones (yellow), initial belief (light blue). (b) and (c) Navigation2: landmarks (pink), goal states (green), obstacles that are present in both the initial map and the modified environment (black), additional obstacles that were not known in the initial map (grey), initial belief (blue). The map in (b) represents the environment used to generate the A* policy. The map in (c) represents the environment used for planning and execution. Navigation3: 3D navigation scenario with uneven terrain, goal area (red).

### 6.1.1 Navigation1: Long Horizon

The environment is a $60 \times 60$ static gridworld populated with obstacles, danger zones, and landmarks, as illustrated in Figure 1a. Details of the actions, observations and reward are presented in Section A.7.1.

POMCP's rollout policy and REFSOLVER's fully observed policy are the A* policy for the fully observed and deterministic version of the problem. POMCP was run with an exploration constant of 300 and maximum depth of 180 – this maximum depth is the upper bound for (tree depth + rollout steps). For REFSOLVER, the maximum tree depth was 90, the maximum rollout depth was 180 and $\alpha = 0.5$. Both methods were executed for a maximum of 180 steps.

### 6.1.2 Navigation2: Slightly Perturbed Environment

The environment is a $60 \times 60$ gridworld populated with obstacles and landmarks. The robot is provided with an initial environment map that was slightly different from the environment used during execution. Details of the scenario (actions, observations, and reward) are presented in Section A.7.2.

For testing purposes, we generated 64 different initial maps with randomly placed obstacles and landmarks. For each initial map, we randomly generate four slight modifications of the initial map by adding obstacles at random. An illustration of an initial environment map and a corresponding environment used during execution are illustrated in Figure 1b and 1c.

Both the POMCP rollout policy and REFSOLVER's fully observed policy are the A* policy for the fully observed and deterministic version of the robot operating in the *initial* environment map. However, the dynamics, observation, and reward that the robot experiences and receives during execution follow the deformed map, which is different from the initial map.

POMCP was run with an exploration constant of 300 and maximum depth of 60 – this maximum depth is the upper bound for (tree depth + rollout steps). For REFSOLVER, the maximum tree depth was 30, the maximum rollout depth was 60 and $\alpha = 0.5$.

### 6.1.3 Navigation3: 3D Navigation Environment

Navigation3 is a 3D environment where a drone has to fly over uneven terrain. The drone starts at the location illustrated in Figure 1d and needs to reach the red goal region, which is located behind the hill. State and observation spaces are continuous, while actions are discrete. See Appendix Section A.7.3 for details.

The rollout and fully observed policy recommend the action that minimises the Euclidean distance between the current state and goal location with no consideration of obstructing terrain. POMCP was run with an exploration constant of 400 and a maximum depth of 150 (upper bound for tree depth + rollout depth). For REFSOLVER, the maximum tree depth was 150, the maximum rollout depth was 200, and $\alpha = 0.4$. Both methods were executed for a maximum of 200 steps.

### 6.2 Performance Comparison

To evaluate the performance on REFSOLVER, we ran each method on each problem scenario 256 times. For each scenario, each method was given 30 seconds planning time per step. The initial policy generation used by POMCP for rollout and by REFSOLVERfor the 2D experiments took an average of $45$ minutes for each scenario. We only need to compute this initial policy once for Navigation1 and as many as the number of initial maps for Navigation2.

For Navigation3, the initial policy is generated on-line and takes a small fraction of the planning time per step. The initial policy is derived from a simple shortest path distance policy that assumes the environment is totally empty, allowing it to be computed fast. Note that this way of generating the initial policy substantially alleviates the

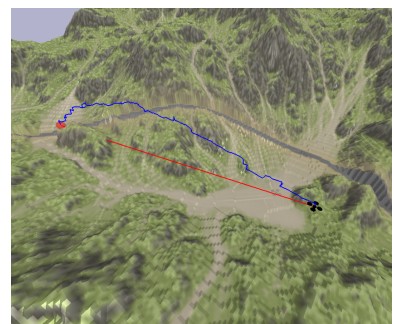

Figure 2: The red line indicates the shortest path from the initial to the goal positions, while the blue line indicates a trajectory trace of executing the policy generated by REFSOLVER.

computational cost, especially for complex problems. The quality of such an initial policy is of course relatively low, as indicated by the red-line in Figure 2. However, our results indicate that REFSOLVER is able to iteratively deform the initial policy to generate a good strategy for the environment the robot operates in. Note that the trace of the deformed policy indicates that the agent is aware of the potential collision danger quite early, even though it requires more than 100 steps lookahead.

Table 1 compares the success rates and the average total discounted reward (with a 95% confidence interval) achieved by POMCP and REFSOLVER, respectively. To better understand the performance improvement with respect to the planning time, we also ran Navigation2 with different planning times per step. The results of 256 runs are presented in Table 2. The results indicate that in all instances and for all planning times, the basic implementation of REFSOLVER was able to reach the goal with a significantly greater likelihood and average total discounted reward.

One may be discouraged by the success rate presented. However, all the problem scenarios presented here have a horizon of over 60 steps, which remains a tremendous challenge for many general approximate on-line POMDP solvers today, as indicated by the performance of POMCP.

Table 1: POMCP vs REFSOLVER with 30 sec planning time per step

| | Navigation1 | | Navigation2 | | Navigation3 | |
| --- | --- | --- | --- | --- | --- | --- |
| Solver | % Succ. | Avg Tot Dis Rw | % Succ. | Avg Tot Dis Rw | % Succ. | Avg Tot Dis Rw |
| POMCP | 1 | $-284 \pm 5$ | 11 | $-181 \pm 13$ | 0 | $-87 \pm 0$ |
| REFSOLVER | 31 | $-201 \pm 24$ | 49 | $-95 \pm 16$ | 64 | $-5 \pm 21$ |

Table 2: POMCP vs REFSOLVER in Navigation2 with different planning time per step.

| | Planning Time Per Step (Seconds) | | | | | |
| --- | --- | --- | --- | --- | --- | --- |
| | 10 | | 20 | | 30 | |
| Solver | % Succ. | Avg Tot Dis Rw | % Succ. | Avg Tot Disc Rw | % Succ. | Avg Tot Disc Rw |
| POMCP | 2 | $-205 \pm 7$ | 1 | $-208 \pm 4$ | 11 | $-181 \pm 13$ |
| REFSOLVER | 30 | $-137 \pm 15$ | 36 | $-128 \pm 15$ | 49 | $-95 \pm 16$ |

# 7 Related Work

## 7.1 MDPs with KL Divergence Minimisation and Reinforcement Learning

The idea of using KL divergence minimisation in fully observable environments goes back to a series of works on *Linearly Solvable MDPs*[2] [24, 25, 26, 2] who used it to efficiently approximate solutions to certain classes of (fully-controllable) MDPs. A learning-based algorithm (*Z-Learning*) was also proposed for problems where the model was not available [24]. Azar et al. [1] extended the approach to general MDPs with stochastic policies and introduced *Dynamic Policy Programming* which converges asymptotically to the optimal (stochastic) policy of the MDP. The authors also demonstrated that the formulation could be suitably adapted using basis functions to deal with large-scale (continuous) state-action problems. A unifying perspective on this approach was then offered by Rawlik et al. [19] who demonstrated that the stochastic control formulation can be related to approximate inference control in trajectory optimisation [28] and provided further experimental validation for a $Z$-Learning-like algorithm. Finally, we note that the literature on KL divergence minimisation is closely related to Maximum Entropy RL where some recent theoretical developments [3] have demonstrated robustness to the model (i.e. dynamics and rewards) in fully observable domains.

---

[2]See Section A.1 for a brief summary of the theory.

As far as the authors are aware, all of the above papers have no parallel in partially-observable environments as of yet. As such, this paper can be seen as first step to extending the above results to such environments.

## 7.2 Sampling-Based Approximate POMDP Solvers

Methods for finding exact solutions to POMDPs [16] are impractical for realistic robotics problems due to their computational complexity. In recent decades, sampling-based approximate methods have significantly scaled up the capabilities of both online and offline POMDP solvers [18, 11, 21, 29, 12] (see [9] for a comprehensive survey). A major constraint in all state-of-the-art methods is that they rely on a dynamic programming paradigm which necessitates enumeration of actions at every belief node of the search tree. As such, existing solvers cannot effectively deal with the curse of history because they do not have sufficient computational resources to perform a sufficiently long look-ahead.

To deal with this problem, methods that abstract the problem using *macro-actions* have been proposed [23, 5, 13, 10] where the agent only plans with respect to judiciously chosen *sequences* of actions, called macro actions, or sub-policies. They reduce the effective planning horizon, but can be sensitive to the class of macro-actions or sub-policy chosen. In contrast, we propose to alleviate the issue via a slight reformulation of the POMDP, which allows numerical computation to infer a close to optimal policy via sparse sampling of both the action and observation spaces. Moreover, existing techniques that reduce the effective planning horizon can also be applied to Reference-Based POMDPs, which will further improve its ability to compute good solutions to POMDPs with long planning horizons.

## 8 Discussion and Further Work

We have introduced the Reference-Based POMDP, which is a modification of the standard POMDP. This modified formulation enables the problem of computing the optimal policy to be analytically transformed into a problem of computing expectations, which can be approximated without exhaustive enumeration of the actions. We showed that, under mild assumptions, the standard POMDP can be related to a Reference-Based POMDP via an embedding (Proposition 4.1). We also presented a preliminary solver REFSOLVER which solves a Reference-Based POMDP. Initial performance comparisons on 2D and 3D navigation problems indicate that this preliminary method is promising and can be employed to substantially outperform state-of-the-art solvers in long-horizon problems.

By introducing KL minimisation with respect to reference probabilities, Reference-Based POMDPs are likely to be suitable for problems with dynamically changing POMDP models with long planning horizons where one needs to deform the policy being followed without making abrupt changes to the policy. We also see applications of this kind of framework in the context of artificial agents that are guided by a human to make responsible decisions while also judiciously transgressing guidance in the case of significant human error.

One limitation of this paper is that the formulation of a Reference-Based POMDP asserts that arbitrarly "belief-to-belief" transitions can be realised which is a remnant of following the approach set out in Todorov [24] too closely.[3] This assumption can be relaxed by stating the Bellman equation with respect to stochastic policies over *actions* in a manner similar to Azar et al. [1] rather than abstract *belief-to-belief* transitions. Such an approach yields very similar analytical results and avoids the need to recover approximate stochastic actions as set out in Section 3.3. We expect to set out these results in a subsequent paper.

The above limitation is just one of the many possible directions for expanding this work. We hope this paper opens new avenues to further scale the capability of POMDP solvers.

## 9 Acknowledgements

This work is done in collaboration with Safran Electronics & Defense Australia Pty Ltd and Safran Group under the ARC Linkage project LP200301612.

---

[3]See [19, 1] for further commentary.

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

# A  Appendix

## A.1  Background on Linearly Solvable MDP

Since the Reference-Based POMDP expands the Linearly Solvable (fully observed) MDPs[24, 25, 26] to POMDPs, for completeness, here, we summarise Linearly Solvable MDPs.

A standard infinite horizon MDP is specified by tuple $\langle \mathcal{S}, \mathcal{A}, \mathcal{T}, R, \gamma \rangle$, where $\mathcal{S}$ and $\mathcal{A}$ are the state and action spaces, $\mathcal{T}(s, a, s')$ is the conditional probability function that specifies the probability the agent arrives at state $s' \in \mathcal{S}$ after performing action $a \in \mathcal{A}$ at state $s \in \mathcal{S}$, $R$ is the reward function, and $\gamma \in (0, 1)$ is the discount factor. The solution to an MDP problem is a an optimal policy $\pi^* : \mathcal{S} \to \mathcal{A}$ that maximises the value function:

$$V^*(s) = \max_{a \in \mathcal{A}} \left[ R(s, a) + \gamma \sum_{s' \in \mathcal{S}} \mathcal{T}(s, a, s') V^*(s') \right] \tag{24}$$

The works in [24, 25, 26] consider a class of MDPs where, the state space $\mathcal{S}$ is finite and for any states $s, s' \in \mathcal{S}$, there exists a one-step (not necessarily time-homogeneous) transition probability $p(s' \,|\, s)$ representing the *passive dynamics* of the system. They propose a new formulation of MDPs, called Linearly Solvable MDPs, to be specified by $\langle \mathcal{S}, p, r, \gamma \rangle$, where $r : \mathcal{S} \to \mathbb{R}$ is the reward function. A solution to the Linearly Solvable MDP is a stochastic state-to-state transition probability $u(\cdot \,|\, s)$ that maximises:

$$v(s) = \sup_{u(\cdot \,|\, s) \in \mathscr{U}_p(s)} \left( r(s) - \mathrm{KL} \left( u(\cdot \,|\, s) \,\|\, p(\cdot \,|\, s) \right) + \gamma \sum_{s' \in \mathcal{S}} u(s' \,|\, s) v(s') \right) \tag{25}$$

where $\mathscr{U}_p(s)$ is the set of admissible controls. An admissible control $u(\cdot \,|\, s)$ is one that prohibits state transitions that are not feasible under the passive dynamics $p(\cdot \,|\, s)$.

Now, suppose $w(s) := e^{v(s)}$ for any $s \in \mathcal{S}$, then (25) is equivalent to

$$w(s) = e^{r(s)} \sum_{s' \in \mathcal{S}} p(s' \,|\, s) w^{\gamma}(s'). \tag{26}$$

Moreover, the solution $w^*$ to the above equation exists and is unique. The optimal stochastic transition to the equation (25) is given by

$$u^*(\cdot \,|\, s) = \frac{p(\cdot \,|\, s) w^{*\gamma}(\cdot)}{D[w^{*\gamma}](s)}. \tag{27}$$

where $D[w^{*\gamma}](s) := \sum_{s' \in \mathcal{S}} p(s' \,|\, s) w^{*\gamma}(s')$ is a normaliser. Intuitively, one can view $w^*$ as the desirability score, so that (27) represents distorting the passive dynamics to transition dynamics that favour transitioning to states with higher desirability scores. Of course, $w^*$ is not known a priori but it can be determined by iterating the Bellman backup operator given by (26). This computation essentially reduces to taking expectations under the reference dynamics, which can be computed faster than searching for the optimal value function in (24) directly.

A standard MDP can be *embedded* in a linearly solvable MDP. This implies that, for a given standard MDP problem $\langle \mathcal{S}, \mathcal{A}, \mathcal{T}, R, \gamma \rangle$, one can embed it as an instance of a linearly solvable MDP, use the above efficient machinery to determine the solution to the linearly solvable MDP $u^*(\cdot \,|\, s)$, and then choose the symbolic action $a^* \in \mathcal{A}$ such that $\mathcal{T}(s' \,|\, s, a^*)$ is as close as possible to $u^*(\cdot \,|\, s)$. Empirical results in [25] indicate that there is a close correspondence between the optimal value of the embedded standard MDP $\langle \mathcal{S}, \mathcal{A}, \mathcal{T}, R, \gamma \rangle$ and the optimal value of the linearly solvable MDP.

## A.2  Proof of Lemma 3.1

*Step 1.* We first need to verify that a maximiser to the supremum in (9) exists. To this end, define $\mathcal{W}(b) := e^{\mathcal{V}(b)}$ for any $b \in \mathcal{B}$ and notice that the terms inside the supremum in the RHS of equation

(9) can be rewritten as

$$\sum_{a,o} U(a,o\,|\,b)\Big[R(b,a) - \log\Big\{\frac{U(a,o\,|\,b)}{\bar{U}(a,o\,|\,b)}\Big\}$$

$$+ \gamma\sum_{a,o}\mathcal{V}\big(\tau(b,a,o)\big)\Big]$$

$$= -\sum_{a,o} U(a,o\,|\,b)\Big[\log\Big\{\frac{U(a,o\,|\,b)}{\bar{U}(a,o\,|\,b)e^{R(b,a)}\mathcal{W}^\gamma(\tau(b,a,o))}\Big\}\Big]$$

$$= -\sum_{a,o} U(a,o\,|\,b)\Big[\log\Big\{\frac{U(a,o\,|\,b)\mathcal{D}[\mathcal{W}^\gamma](b)}{\bar{U}(a,o\,|\,b)e^{R(b,a)}\mathcal{W}^\gamma(\tau(b,a,o))}\Big\}$$

$$- \log\big\{\mathcal{D}[\mathcal{W}^\gamma](b)\big\}\Big]$$

$$= -\operatorname{KL}\Big(U(\cdot,\cdot\,|\,b)\,\Big\|\,\frac{\bar{U}(\cdot,\cdot\,|\,b)e^{R(b,a)}\mathcal{W}^\gamma\big(\tau(b,a,o)\big)}{\mathcal{D}[\mathcal{W}^\gamma](b)]}\Big)$$

$$+ \log\big\{\mathcal{D}[\mathcal{W}^\gamma](b)\big\} \quad (28)$$

where $\mathcal{D}[\mathcal{W}^\gamma](b) := \sum_{a,o}\bar{U}(a,o\,|\,b)e^{R(b,a)}\mathcal{W}^\gamma\big(\tau(b,a,o)\big)$ is a normalising factor. Only the KL divergence term in the last line above depends on $U$. We know that the KL divergence is minimised when its two component distributions are identical. That is, when

$$U^*(a,o\,|\,b) = \frac{\bar{U}(a,o\,|\,b)e^{R(b,a)}\mathcal{W}^\gamma\big(\tau(b,a,o)\big)}{\mathcal{D}[\mathcal{W}^\gamma](b)}. \quad (29)$$

It is clear that $U^*$ belongs to the space $\mathscr{U}(b)$ since $\bar{U}(a,o\,|\,b) = 0$ implies that $U^*(a,o\,|\,b) = 0$ too. Therefore, we conclude that the supremum is attained and that $U^*$ is the maximiser.

*Step 2.* Now, we can essentially repeat the classical argument from Ross [20] (see e.g. Theorem 6.5). Namely, let $\Phi : \mathbb{B}(\mathcal{B}) \to \mathbb{B}(\mathcal{B})$ be the Bellman backup operator

$$\Phi\mathcal{V}(b) := \sup_{U\in\mathscr{U}(b)}\Big(R(b,U) - \operatorname{KL}(U\,\|\,\bar{U}) + \gamma\,\mathbb{E}_U\big[\mathcal{V}\big(\tau,\cdot,\cdot\big)\big]\Big) \quad \forall b\in\mathcal{B} \quad (30)$$

where, for brevity, we write

$$\operatorname{KL}(U\,\|\,\bar{U}) := \operatorname{KL}\big(U(\cdot,\cdot\,|\,b)\,\|\,\bar{U}(\cdot,\cdot\,|\,b)\big) \quad (31)$$

and

$$\mathbb{E}_U\big[\mathcal{V}\big(\tau,\cdot,\cdot\big)\big] := \sum_{a,o} U(a,o\,|\,b)\mathcal{V}\big(\tau(b,a,o)\big). \quad (32)$$

We want to show that $\Phi$ is a contraction. For any $b\in\mathcal{B}$ and any $\mathcal{V}_1,\mathcal{V}_2\in\mathbb{B}(\mathcal{B})$,

$$(\Phi\mathcal{V}_1)(b) - (\Phi\mathcal{V}_2)(b)$$

$$= \sup_{U\in\mathscr{U}(b)}\Big(R(b,U) - \operatorname{KL}(U\,\|\,\bar{U}) + \gamma\,\mathbb{E}_U\big[\mathcal{V}_1\big(\tau,\cdot,\cdot\big)\big]\Big)$$

$$- \sup_{\tilde{U}\in\mathscr{U}(b)}\Big(R(b,\tilde{U}) - \operatorname{KL}(\tilde{U}\,\|\,\bar{U}) + \gamma\,\mathbb{E}_{\tilde{U}}\big[\mathcal{V}_2\big(\tau,\cdot,\cdot\big)\big]\Big)$$

$$\le \Big(R(b,U^*) - \operatorname{KL}(U^*\,\|\,\bar{U}) + \gamma\,\mathbb{E}_{U^*}\big[\mathcal{V}_1\big(\tau,\cdot,\cdot\big)\big]\Big)$$

$$- \Big(R(b,U^*) - \operatorname{KL}(U^*\,\|\,\bar{U}) + \gamma\,\mathbb{E}_{U^*}\big[\mathcal{V}_2\big(\tau,\cdot,\cdot\big)\big]\Big)$$

$$= \gamma\sum_{a,o} U^*(a,o\,|\,b)\Big[\mathcal{V}_1\big(\tau(b,a,o)\big) - \mathcal{V}_2\big(\tau(b,a,o)\big)\Big]$$

$$\le \gamma\,\|\mathcal{V}_1 - \mathcal{V}_2\|_\infty \quad (33)$$

where $U^*$ is the maximiser of

$$R(b,U) - \operatorname{KL}(U\,\|\,\bar{U}) + \gamma\,\mathbb{E}_U\big[\mathcal{V}_1\big(\tau,\cdot,\cdot\big)\big]. \quad (34)$$

Reversing the roles of $\mathcal{V}_1$ and $\mathcal{V}_2$ and using the fact that $b\in\mathcal{B}$ is arbitrary, we conclude that

$$\|\Phi\mathcal{V}_1 - \Phi\mathcal{V}_2\|_\infty \le \gamma\,\|\mathcal{V}_1 - \mathcal{V}_2\|_\infty. \quad (35)$$

Since we assumed that $\gamma\in(0,1)$, we conclude that $\Phi$ is a contraction.

## A.3 Proof of Theorem 3.1

Repeating the argument in Step 1 of A.2, we see that the Bellman equation (9) reduces to

$$\mathcal{V}(b) = \log\big[\mathcal{D}[w^\gamma](b)\big] = \log\Big[\sum_{a,o}\bar{U}(a,o\,|\,b)e^{R(b,a)}\mathcal{W}^\gamma\big(\tau(b,a,o)\big)\Big] \tag{36}$$

which, after taking exponents, justifies the equivalence to (11). Given this equivalence and Lemma 3.1, it is clear that (11) has a unique solution. To be more explicit, suppose for a contradiction that (11) does not have exactly one solution (up to $\|\cdot\|$-equivalence of solutions). Then by the equivalence between the two Bellman equations, (9) would either have no solutions or more than one solution which contradicts the existence and uniqueness guaranteed by Lemma 3.1. Finally, (12) follows from the form of the maximiser at each Bellman step.

## A.4 Proof of Proposition 3.1

For brevity, we will fix a $b \in \mathcal{B}$ and drop it from our notation. Also write $\pi = \pi(\cdot\,|\,b)$ and $\pi_a = \pi(a\,|\,b)$. The Lagrangian for the constrained problem is

$$\mathcal{L}(\pi,\lambda) = \sum_{a,o} P(o\,|\,a)\pi_a \log\Big[\frac{P(o\,|\,a)\pi_a}{U^*(a,o)}\Big] + \lambda\Big(\sum_a \pi_a - 1\Big). \tag{37}$$

We require, in addition, that the minimiser $\pi^*$ (which exists due to the Weierstrass extreme value theorem) is such that $\pi_a^* \geq 0$ for each $a \in \mathcal{A}$. The first order necessary conditions gives

$$\pi_a = e^{-(1+\lambda)}\exp[-\Pi(a)] \quad \forall a \in \mathcal{A} \tag{38}$$

and the constraint equation gives

$$1 = \sum_a \pi_a = e^{-(1+\lambda)}\sum_a \exp[-\Pi(a)]. \tag{39}$$

Hence the only candidate for the minimiser is $\pi^*$ such that

$$\pi_a^* = \frac{\exp[-\Pi(a)]}{\sum_{\hat{a}\in\mathcal{A}}\exp[-\Pi(\hat{a})]} \quad \forall a \in \mathcal{A}. \tag{40}$$

The Hessian of $\mathcal{L}$ is positive definite for any $\lambda$ and $\pi \in \Delta(\mathcal{A})$, so we conclude that $\pi^*$ is a minimiser. Finally, that $\pi_a^* \geq 0$ for every $a \in \mathcal{A}$ is clear from (40).

## A.5 Proof of Proposition 4.1

*Proof.* Fix an $\hat{a} \in \mathcal{A}$ and $b \in \mathcal{B}$. If we set

$$\pi^{\hat{a}}(a\,|\,b) := \begin{cases} 1, & a = \hat{a} \\ 0, & \text{otherwise} \end{cases} \tag{41}$$

and

$$U^{\hat{a}}(a,o\,|\,b) := P(o\,|\,\hat{a},b)\pi^{\hat{a}}(a\,|\,b) \in \mathcal{U} \tag{42}$$

then the constraint (20) is satisfied trivially. Straightforward computations show that the constraint (21) is satisfied by $U^{\hat{a}}(a,o\,|\,b)$ with $(\rho,\bar{U})$ as defined in (22) and (23). $\square$

## A.6 Algorithm REFSOLVER

---

**Algorithm 1** REFSOLVER

---

**parameters:**
  $\langle \mathcal{S}, \mathcal{A}, \mathcal{T}, R, \gamma \rangle$
  max-depth
  max-rollout-depth
  $\alpha$                    ▷ expl const $= 1 - \alpha$
**require:** $\gamma \in (0,1), \alpha \in [0,1)$

---

**PRE-PROCESS (OFFLINE)**

---

1: $\pi^{\text{FO}} \leftarrow$ GENERATE-FO-POLICY($\langle \mathcal{S}, \mathcal{A}, \mathcal{T}, R, \gamma \rangle$)

---

2: **RUNTIME (ONLINE)**

---

3: **procedure** PLAN-AND-EXECUTE($h$)
4:   **repeat**
5:     **if** $h = \emptyset$ **then**
6:       $s \sim \mathcal{I}$
7:     **else**
8:       $s \sim \mathcal{B}(h)$
9:     **end if**
10:     SIMULATE($s, h, 0$)
11:   **until** TIMEOUT()
12:   **return** OPTIMAL-STOCHASTIC-POLICY($h$)
13: **end procedure**

14: **procedure** ROLLOUT($s, h$, depth)
15:   $a \leftarrow \pi^{\text{FO}}(s)$
16:   **if** $s \in \mathcal{G}$ or depth $>$ max-rollout-depth **then**
17:     **return** $R(s,a)$
18:   **end if**
19:   $(s', o, R) \sim \mathcal{G}(s,a)$     ▷ generative model
20:   **return** $R(s,a) +$ ROLLOUT($s', hao$, depth $+$ 1)
21: **end procedure**

22: **procedure** SIMULATE($s, h$, depth)
23:   **if** $s \in \mathcal{G}$ or depth $>$ max-depth **then**
24:     **return** $\exp($ROLLOUT($s, h$, max-depth)$)$
25:   **end if**
26:   $\mathcal{B}(h) \leftarrow \mathcal{B}(h) \cup \{s\}$
27:   $N(h) \leftarrow N(h) + 1$
28:   $X \sim$ Bernoulli($\alpha$)
29:   $a \leftarrow \pi^{\text{FO}}(s)I_{\{X=1\}} +$ Unif-Act() $\times I_{\{X=0\}}$
30:   $(s', o, R) \sim \mathcal{G}(s,a)$
31:   $N(ha) \leftarrow N(ha) + 1$
32:   $\widehat{R}(ha) \leftarrow \widehat{R}(ha) + \frac{R(s,a) - \widehat{R}(ha)}{N(ha)}$
33:   $\widehat{\mathcal{W}} \leftarrow \widehat{\mathcal{W}} + \frac{e^{\widehat{R}(ha)}\text{SIMULATE}(s', hao, \text{depth}+1) - \widehat{\mathcal{W}}(h)}{N(h)}$
34:   **return** $\widehat{\mathcal{W}}(h)^{\gamma}$
35: **end procedure**

36: **procedure** OPTIMAL-STOCHASTIC-POLICY($h$)
37:   $\mathcal{D} \leftarrow 0$                    ▷ Normaliser
38:   **for** $a \in \mathcal{A}$ and $o \in \mathcal{O}$ **do**
39:     **if** $hao \notin T$ **then**
40:       $\widehat{U}^*(hao) = 0$
41:     **else**
42:       $\widehat{U}^*(hao) \leftarrow \frac{N(hao)}{N(h)} e^{\widehat{R}(ha)} \mathcal{W}^{\gamma}(hao)$
43:     **end if**
44:     $\mathcal{D} \leftarrow \mathcal{D} + \widehat{U}^*(hao)$
45:   **end for**
46:   **for** $a \in \mathcal{A}$ **do**
47:     $\Pi(a) \leftarrow \frac{N(hao)}{N(ha)} \log \left[ \frac{N(hao)\mathcal{D}}{N(ha)\widehat{U}^*(hao)} \right]$
48:   **end for**
49:   $\pi^* \leftarrow \{a : \exp[-\Pi(a)]/\mathcal{D}^{\Pi}\}$
50:   **return** RANDOM-SAMPLE($\pi^*$)
51: **end procedure**

---

## A.7 Details of Experimental Scenarios

### A.7.1 Details of Navigation1 Scenario

The robot can move in the four cardinal directions with 0.1 probability of actuator failure. If the realised movement leads to a collision with an obstacle or the edge of the map, no movement occurs and the robot remains in its current position. If the robot's true state is a landmark, the robot receives a position reading uniformly in the $9 \times 9$ grid around the robot's true state. Outside the landmarks, the robot receives no observation. The robot receives a penalty of -100 for entering a danger zone and a reward of +300 for entering a goal state. In both cases, the problem terminates. Every other state incurs a reward of -1. The discount parameter was 0.99. The robot's initial belief was equally distributed between two initial positions that were uniformly sampled from the southern-most row of the map.

### A.7.2 Details of Navigation2 Scenario

Similar to Navigation1, the robot's action space consists of moves anywhere in the four cardinal directions NORTH, SOUTH, EAST, WEST. To simulate noise in the robot's actuator's, actions fail with 0.1 probability, and if this occurs, the robot moves randomly in a direction orthogonal to the one specified. If the realised movement leads to a collision with an obstacle or the edge of the

map, no movement occurs and the robot remains in its current position. If the robot's true state is a landmark, the robot receives a position reading uniformly in the $9 \times 9$ grid around the robot's true state. Otherwise, the robot receives no observation. The robot receives a reward of +600 for being in a goal state, and -3 for being in any other state. The discount parameter was $\gamma = 0.99$. The robot's initial belief was equally distributed between two initial positions that were uniformly sampled from the southern-most row of the map.

### A.7.3   Details of Navigation3 Scenario

The state and observation spaces have six dimensions which represent the position $(x, y, z)$ and orientation (roll, pitch, yaw) of the drone. There are 12 actions that increase or decrease one dimension by 0.2 units. Action and observation noise is sampled from a multivariate Gaussian distribution with $0$ mean and $0.032$ standard deviation. If an action results in a collision with the terrain, the robot receives a -500 penalty, and the world terminates. Reaching the goal area receives a 1000 reward, and each step incurs a -1 penalty. The drone's initial configuration is $(-4.0, -1.0, 0.0, 0.0, 0.0, 0.0)$ and the goal are is the ball with centre $(10.4, 3.0, 0.45)$ and radius $0.25$. The start and goal are not in a direct line and the robot needs to go over the hill near the goal area to reach its destination. The discount parameter was $\gamma = 0.99$. The robot's initial belief is populated with its exact start position.

### A.8   Source Code

We include the source code for REFSOLVER, which is developed on top of pomdp_py as a Supplementary Material. `https://github.com/RDLLab/ref_pomdp_neurips23`

