# OpenReview forum: "Reference-Based POMDPs"
_NeurIPS.cc/2023/Conference — NeurIPS 2023 poster_

### Official Review · Reviewer_Hi3x · 2023-07-04

**Soundness:** 3 good
**Presentation:** 3 good
**Contribution:** 2 fair
**Rating:** 5
**Confidence:** 3

**Summary:**

This paper presents a method to solve POMDPs by considering a reference policy. The solution of a reference-based POMDP is presented. Then the existence and uniqueness of the solution are proved. Then the author shows the connections between a reference-based POMDP and a standard POMDP. In the end, an online planning algorithm is developed.

**Strengths:**

The POMDP problem is worth exploring, starting from the case that a reference policy is available is a good start.

The general idea is easy to follow.

Theorems are developed for reference-based POMDP. Although some steps rely on approximation, the logic chain of the analysis is complete.

**Weaknesses:**

It is a bit hard to understand all the math details. It is unclear to me what is the purpose of theorem 3.1.

How to get the initial reference policy can be a major barrier to applying this method. For example, the author mentioned a way to get the reference policy from a fully observable environment seems to be a leak of the optimal solution, which I think is inappropriate.

Please present the assumptions explicitly.

The experiment is relatively simple and not convincing enough. It is unfair to compare the baseline with the method when the reference policy is derived from a fully observable environment.

**Questions:**

Does the reference policy always exist and available?

What's the purpose of theorem 3.1?

When a standard POMDP can not be approximated by a reference-based POMDP?

Is there any other baseline that came up in the past 13 years? The baseline seems to be out of date.



**Limitations:**

A reference policy must be available.

---

> ### Author Rebuttal · Authors · 2023-08-08
>
> Thank you for the positive review and for your questions.
>
> Re “Weaknesses: purpose of theorem 3.1”. This is the main Theorem in this paper and is a straightforward extension of the LS-MDP of Todorov (see eq 31 and 32) to the POMDP. It’s main point is to demonstrate that: 1) the Bellman equation for a Ref-POMDP can be represented as an expectation under the passive dynamics
> ; 2) the optimal policy (by definition, a belief-to-belief transition probability) can be obtained explicitly after solving the Bellman equation. As noted in the paper, these facts can be leveraged to efficiently approximate the solution.
>
> Re ”Weaknesses, how to get the initial reference policy…” We do acknowledge that a judicial choice for the reference policy needs to be made as the problem formulation, by definition, rewards respecting the reference policy (see eq 10). However, a solution to a Ref-POMDP will also ignore the reference policy if greater rewards are available by deviating from it (also eq 10). As such, there is a level of robustness against the reference policy baked in to the formulation. A good way to think of the reference policy is as a “heuristic” or “initial guess” which can be jettisoned by the Ref-POMDP’s solution if it is advantageous to do so. Note also, in our environments, the reference policy generated by the A* policy is suboptimal in general due to partial observability and transition noise (e.g. starting at S1 in Navigation 1 and proceeding via the A* policy almost certainly leads to very negative results). As noted in the summary, we envisage RefSolver to be suitable for problems with dynamically changing POMDP models where one needs to deform the policy being followed without making abrupt changes to the policy. We also hope that for more general problems we can substantially revise the algorithm by iteratively improving the reference policy.
>
> As a general point, note that most complex problems in optimisation require a heuristic or initial policy that can be iteratively improved so we do not believe that such the requirement for a reference policy is too restrictive.  Generally, as we suspect it is the case here, the proximity of the “guess” to the final solution also determines the speed of convergence.
>
> Re "Please present the assumptions explicitly." Agreed that we can state assumptions and scope of application clearly upfront. This can be updated in the final version.  The key assumptions of our formulation is that a stochastic belief-to-belief transition reference policy needs to be given and that the model is known.
>
> Re ”Weaknesses, the experiment is relatively simple…” We acknowledge that the experiments are preliminary, but they demonstrate some key advantages of our approach. Both navigation problems have a long horizon, and we demonstrate in Navigation 2 that the modified environment can be successfully navigated even when the A* policy is generated off the initial map (i.e. A* policy is suboptimal). This is one application of Ref POMDPs which we think could be leveraged in applications where there is some uncertainty about the real environment. Note also that, while we used a fully observable policy as a reference here, this is not a requirement to employ the formulation (e.g. an offline POMDP policy from a similar environment could be used).
>
> Re ”Q, “When a standard POMDP can not be approx….” See lines 230 – 234. Essentially the result says that the representation of the belief space needs to be simple.   More explicitly, the size of the representation should be small with respect to the number of states.
>
> Re "Q, is there any other baseline..." While POMCP and DESPOT are relatively old algorithms, they still represent the canonical benchmarks to beat for online POMDP solvers especially for discrete models. See reference [9] for a recent survey of the field and various extensions of these main algorithms.

---

> > ### Comment · Reviewer_Hi3x · 2023-08-11
> >
> > Thanks for the clarification.

---

### Official Review · Reviewer_Kh8M · 2023-07-06

**Soundness:** 3 good
**Presentation:** 2 fair
**Contribution:** 3 good
**Rating:** 6
**Confidence:** 5

**Summary:**

The paper proposes to regularize online policy search in POMDP by providing a reference stochastic policy. The idea is illustrated on two synthetic grid domains.

**Strengths:**

The paper attempts to systematically incorporate prior knowledge to improve online POMDP planning.

**Weaknesses:**

1) The proposed approach is not well placed in the literature. The approach is not new --- the reference policy is what is known as policy prior in Bayesian approach or policy regularizer in frequentist approach. There are quite some works on that. If the work provides a new formalism to these notions, this should be properly related to prior work.

2) The empirical evaluation is insufficient. The approach is evaluated on two synthetic grid worlds only, on which it outperformed some implementations of POMCP, and was not compared with DESPOT. It is easy to craft problem instances for any algorithm so that the algorithm looks the best, this is a fact known as 'no free breakfast theorem' in computer science. For an informative evaluation, the algorithm should be compared on domains and settings on which POMCP and DESPOT were compared.

**Questions:**

How is the proposed approach related to methods developed for stochastic control with latents? Seems to be the same setting.

**Limitations:**

The limitations are addressed adequately.

---

> ### Author Rebuttal · Authors · 2023-08-08
>
> Thank you for taking the time to provide feedback. See responses below.
>
> On “ The approach is not new – the reference policy is what is known …” We respectfully disagree.
>
> First, we would like the clarify the main contributions of the paper by referring you to the general comments of our rebuttal above.
>
>  Second, we like to note that although at the very high level, the mechanism may seem like a policy prior in a Bayesian approach, the reformulation and implication of the reformulation to solving POMDPs is, to the best of our knowledge, novel and distinct to existing approaches.   We emphasise that the major contribution of the paper is that we propose a reformulation of POMDPs, such that the  optimisation for solving POMDPs no longer requires exhaustive enumeration of the action space (this enumeration is required by almost any POMDP planning method today). Instead, optimisation can be performed analytically as a consequence of the relaxation which, under suitable transformations, leads to a linearisation of the Bellman equation (Theorem 3.1) and allows the Bellman equation to be computed as an expectation, which in turn can be more efficiently computed using Monte Carlo methods relative to other state-of-the-art POMDP solvers.
>
> This contribution represents an extension of the theory of Linearly Solvable MDPs (see Todorov [18-20]) to partially observable domains. Note that Linearly Solvable MDPs are closely related to Max Ent RL as noted in Section 7 Summary of the paper, where some recent theoretical developments have demonstrated robustness to the model (i.e. dynamics and rewards) in fully observable domains (see [2]) but again we are not aware of a parallel in partially-observable domains as of yet.  Of course, model uncertainty could be explored at a later stage and is important for real-life applications, but this paper's scope was simply to demonstrate both theoretically and experimentally the feasibility of Ref POMDPs in the context of partial observability where the model itself (i.e. transition, observation and rewards) are known.
>
> Re Weakness 2. We agree that further experiment on higher-dimensional environments should be done to further validate this approach. However, it should be noted that we have not manufactured problems to yield good results. E.g. If the A* policy were blindly observed in Navigation1, this would consistently yield poor results. What both our experiments demonstrate is that RefSolver can improve on “good” (albeit suboptimal) reference policies which is the main point. We did not compare RefSolver to DESPOT due to the long-horizon environments as stated in the paper (see lines 264-267). Constructing a DESPOT for 4 actions with a max depth of 60 is very expensive because each action needs to be visited at each observation branch. This issue has also been highlighted in [5]. However, we do agree that it would be appropriate to run experiments in higher-dimensional environments with shorter planning horizons.
>
> Re Questions: Please refer to our clarification on the main contribution in the general comment.

---

> > ### Comment · Reviewer_Kh8M · 2023-08-12
> >
> > Thank you for your response.  Based on the response, I still think proper discussion of relation of this method to optimal stochastic control theory and practice is related, as well as much more thorough empirical evaluation. It seems that you agree with both points.

---

> > > ### Author Response · Authors · 2023-08-15
> > > **The paper's contribution is theoretical rather than empirical**
> > >
> > > On the point of a discussion of the literature, we point out again that the inspiration of this paper was the framework of the Linearly-Solvable MDP (see references to papers by Todorov [1, 18-20]). This Linearly-Solvable MDP work is summarised in 8.1 of the paper. We can of course expand this summary to include a broader optimal stochastic control and will do so in the final manuscript if the paper is accepted.
> > >
> > >
> > > On the point of empirical evaluation, we note that the main focus of this work was theoretical rather than experimental, our focus being to establish a well-grounded formulation of Reference-Based POMDPs and a thorough analysis of the features of the problem. This, by itself, is a novel contribution. Related to this point, we also like to highlight three points:
> > >
> > > 1/ An inspiration of this work was Todorov (2006) Linearly-solvable Markov Decision Processes (LS-MDPs), which was published in Adv. in NeurIPS, where the theory of LS-MDPs is developed in a fully-observable context.  Todorov’s NeurIPS paper contribution was theoretical.  At the time, empirical results, while encouraging and supportive of the theory, were preliminary and limitations remained to be clarified, but further investigations demonstrated that this was a fruitful approach.
> > >
> > > 2/ We stress again that the experiment we have run is not trivial due to the long planning horizon and the challenges of partial state observability and state-transition uncertainty. In fact, the environments were chosen quite deliberately to highlight these features and are difficult to navigate even for state-of-the-art solvers such as POMCP, or DESPOT.
> > >
> > > 3/ We did not tune the environment in our experiments to favour our method, and in fact our environments are not well suited for the A* reference policy either.
> > >
> > >
> > > The contribution of RefSolver is that it can improve on a policy, including a less suitable reference policy, by sampling judiciously using our analytical solution in Theorem 3.2.  Crucially, as a well-justified form of Monte Carlo action sampling is used, RefSolver can avoid exhaustive enumeration over actions which is a major limitation for current online methods for long-horizon POMDP problems.  We hope that the reviewer can recognise this contribution and the challenges that are inherent to the formation of the theory here.

---

> > > > ### Comment · Reviewer_Kh8M · 2023-08-15
> > > > **convinced**
> > > >
> > > > you convinced me that the paper has merits sufficient for acceptance. I urge you to update the paper based on the discussion if the paper is accepted.

---

> > > > > ### Author Response · Authors · 2023-08-20
> > > > >
> > > > > Thank you and yes, we will update the paper accordingly as per the feedback and discussion here.

---

### Official Review · Reviewer_RFcX · 2023-07-06

**Soundness:** 3 good
**Presentation:** 2 fair
**Contribution:** 2 fair
**Rating:** 5
**Confidence:** 2

**Summary:**

The authors propose reference based POMDPs, where the agent needs tradeoff between achieving environment rewards and following a reference policy.

**Strengths:**

generally, I think the reference based pomdps are a good idea and can solve some real world problems;

**Weaknesses:**

see below

**Questions:**

1 As for the definition of the policy, is that $\pi(a|b)$ or $\pi(a,o|b)$? In L143 it seems to be the latter. Then, is the belief-to-belief transition $U(a,o|b)$ equals to a policy ? However, the $p(o|a,b)$ seems to contain the environment observation probability which is the intrinsic nature of the environment. As for the reference policy, is it correct to say the reference based pomdps only requires it to provide $p(a|b)$ as in Eq.6? I think this part is somehow confusing for me and I suggest further clarification.

2 According to Eq 10 and Eq 19, the policy chooses sup $U$ while the standard POMDP chooses max $a$. can you further explain the differences between the two? Is it correct to see the reference based POMDP equals to a standard POMDP with different reward function?

3 In section 5 and experiments, the reference policy mainly restricted to a fully-observed based policy. I think reference-based pomdp setting pretty much relies on the performance of the reference policy, and in real world, the base policy might be much worse since the fully observation might not be accessible. It will be interesting to see whether the methods can run well with different initial policy, especially when the initial policy is not that good. Can the methods still benefit from it?

---

> ### Author Rebuttal · Authors · 2023-08-08
>
> Thank you very much for your review and feedback.
>
> Re Question 1. It should be $\pi(a, o | b)$ or equivalently $\pi(b’ | b)$.  By definition of the problem, one chooses belief-to-belief distributions over next beliefs (or, equivalently, action-observation pairs). This is a relaxation of the concept of picking a single action (i.e. point mass) as is classical. A policy is a mapping from a given belief to a distribution over action-observation pairs.
>
> At the most abstract level the reference policy could be given in the top-down form (5) – e.g. a noisy version of an offline POMDP policy computed on a similar environment.  However, in some cases it might be more appropriate to construct the reference policy from the ground up as in (6) and (7).  We acknowledge that the notation in $U^p$ in (5) is a little misleading as it seems to assume that some reference $p$ is given a priori.  This could be changed in the final paper e.g. $\bar{U}$ to denote a top-down policy versus $U^p$ to denote a policy generated from the ground-up.
>
> Re Question 2. We stress again that the two problem formulations are different but related.  The main difference is that a Ref POMDP tries to balance trading off respecting a reference policy while also achieving higher rewards - see (10) while a standard POMDP does not. As noted in the general comments and the paper, this leads to computational advantages which can be exploited to deal with standard POMDPs.
>
> The reason for the supremum is that  a Ref POMDP requires choosing a distribution over action-observation pairs rather than a specific action. As the space of distributions is infinite-dimensional and objective functions are only assumed to be bounded, a supremum is taken. It turns out that the maximiser is actually attained (see Step 1 in Proof of Lemma 3.1) but this is not known a priori.
>
> Moreover, the Bellman equations differ so Ref POMDPs and classical POMDPs really are not the same thing. Compare (2) and (10). However, we demonstrate in Section 4 that a classical POMDP can be embedded inside a Ref POMDP provided the reachable belief space is not too complex. This means that one could approximate solutions to classical POMDPs by casting them as Ref POMDPs. Because solving RefSolver is innately faster than standard POMDPs, we could take advantage of this idea to find efficient solvers for some POMDPs. We are currently further investigating this idea.
>
> Re Question 3. We do acknowledge that a judicial choice for the reference policy needs to be made as the problem formulation, by definition, rewards respecting the reference policy (see eq 10). Of course, if the initial reference policy is very misleading, then this would not lead to useful solutions for the POMDP. However, a solution to a Ref-POMDP will also ignore the reference policy if greater rewards are available by deviating from it (also eq 10). As such, there is a level of robustness against the reference policy baked in to the formulation. A good way to think of the reference policy is as a “heuristic” or “initial guess” which can be jettisoned by the Ref-POMDP’s solution if it is advantageous to do so. As noted in the summary, we envisage RefSolver to be suitable for problems with dynamically changing POMDP models where one needs to deform the policy being followed without making abrupt changes to the policy. We also hope that for more general problems we can substantially revise the algorithm by iteratively improving the reference policy.
>
> As a general point, note that most complex problems in optimisation require a heuristic or initial policy that can be iteratively improved so we do not believe that requiring a reference policy is too restrictive.    Generally, as we suspect it is the case here, the proximity of the “guess” to the final solution also determines the speed of convergence.

---

> > ### Comment · Reviewer_RFcX · 2023-08-19
> >
> > Thanks for your clarification.

---

### Official Review · Reviewer_poha · 2023-07-08

**Soundness:** 3 good
**Presentation:** 3 good
**Contribution:** 3 good
**Rating:** 5
**Confidence:** 3

**Summary:**

The paper introduces and investigates the concept of reference-based POMDP, which addresses the challenge of finding an optimal policy in partially observable environments. The main objective is to simplify this problem by leveraging a baseline fully observed policy. The authors propose that solving a reference-based POMDP can be achieved by iteratively computing expectations using the provided reference (stochastic) policy. They further argue that this approach can be viewed as an extension of Linearly Solvable MDPs to scenarios involving partial observability.

**Strengths:**

- Solving POMDPs is a challenging problem.
- The authors present a new problem formulation for POMDPs called Reference-based POMDPs that uses a reference baseline policy.
- The authors present solution approach to this new problem formulation.

**Weaknesses:**

- It is hard to place the work in current state of the literature in this field because of a lack of discussion on related works.

**Questions:**

1. The paper presents a systematic way to solve POMDPs, albeit with some heavy assumptions on:

    a. availability of a reference belief-to-belief transition probability - a distribution which is constructed using a fully observed policy.

    b. knowledge of admissible reference transition probabilities. computing iterative expectations using the given reference (stochastic) policy.

It would be nice to highlight these assumptions and exact contributions in the introduction.

2. Is  the second paragraph in Introduction just an interpretation of the paper’s formulation/ main algorithm (Alg. 1 included in appendix 8.7)? If it is just an interpretation, the logical flow of paragraph seems conflicting because:

    a. The paragraph starts by saying the problem of reward maximization in POMDPs is challenging. And then proposes to use constrained optimization as a solution - when in general constrained optimization problems are more complex than unconstrained ones.

    b. Since the optimal value function of the Reference-Based POMDP can be computed by purely computing expectations under the reference transition probabilities recursively using Monte Carlo approximation, where exactly is constrained optimization coming into picture?


3. In paragraph 3 in Introduction, authors mention “the standard POMDP can be related to the Reference-Based POMDP via an embedding” - what does an embedding mean here? I would be nice to elaborate this.


4. The paper does not include a section on related works. If it was because of space constraints, I strongly recommend adding this section in Appendix. From reading the paper, it is unclear to me what the state-of-the-art in solving POMDPs is - both algorithms and evaluation environments. For example, the paper does not comment about this recent work [1] on solving POMDPs using approximate information state.


5. For the evaluation environments considered, I believe one could also solve these problems using meta-RL (VariBAD [2]) or robust-RL. Can the authors comment on why is it better to cast these problems as POMDPs as opposed to meta-RL or robust-RL?


I would be happy to increase my score if the authors could clarify the queries above.


References:

[1] Subramanian, J., Sinha, A., Seraj, R., & Mahajan, A. (2022). Approximate information state for approximate planning and reinforcement learning in partially observed systems. *The Journal of Machine Learning Research*, *23*(1), 483-565.

[2] Zintgraf, L., Schulze, S., Lu, C., Feng, L., Igl, M., Shiarlis, K., ... & Whiteson, S. (2021). Varibad: Variational bayes-adaptive deep rl via meta-learning. *The Journal of Machine Learning Research*, *22*(1), 13198-13236.

**Limitations:**

A more detailed discussion on limitation of the solution approach to very high dimensional problems can be included.

---

> ### Author Rebuttal · Authors · 2023-08-08
>
> Thank you very much for your review and feedback.
>
> RE Weaknesses. We will provide additional related work, in relation to Max Entropy RL and stochastic control on top of those already provided. The paper is inspired by an interesting series of papers by Todorov (see ref 18-20), namely Linearly Solvable MDPs, which is summarised in 8.1. These works focused on MDPs whereas we generalise the approach to POMDPs and provide some ways of dealing with the infinite-dimensionality of the belief space using an approximate solver for the Ref POMDP.
>
> RE Question 1. We do not need a reference belief-to-belief transition probability that is constructed using a fully-observed policy. Our fully-observed policy was just one example of a bottom up approach to construct such a policy.  Rather, it is enough to have any initial reference policy which (stochastically) transitions from belief to belief that serves as an initial guess to the problem. The state transition of the reference policy need not be known; only the belief-to-belief level needs to be known.  For instance, the reference policy could be sampled under some generative model.  Some good guesses might be sampled free dynamics or human inputs or an offline POMDP policy from a similar environment.  Of course, as in any optimisation problem, if this initial guess is close to the optimal solution this is useful, but guesses can be iteratively improved upon.
>
> RE Question 2. I wonder if you could clarify what you mean by “constrained optimisation”. For a precise formulation of the modified problem see equation (10). We stress that this is an entirely different problem to the standard POMDP where a "reference transgression term" is introduced by the KL term. It turns out that, under suitable transformations, each Bellman step can be optimised analytically (12) which is why the Bellman equation can be solved by taking pure expectations under the reference measure.  Note that, we do mention “constrained optimisation” in line 185 but this is in the context of the recovering stochastic actions rather than with respect to the formulation.
>
> RE Question 3. See Definition 4.1 and Section 4 for a complete explanation. This was inspired by Todorov’s idea which is briefly summarised in Section 8.1 (lines 403-408).  Intuitively, we say that a standard POMDP can be embedded in a Reference-Based POMDP if we can choose parameters to the Ref-Based POMDP such that, for any action in the standard POMDP, there are points in the relaxed space of stochastic actions where the objective functions align.  The idea is that if an embedding can be found, there is a hope that the solution to the Ref-Based POMDP should be close to that of the standard POMDP and so the more efficient machinery of the Ref-Based POMDP can be readily applied.
>
> RE Question 4. Thank for for this feedback. We provided the background in Appendix (8.1) and will expand the scattered related work into a section in the Appendix too. Briefly regarding state of the art, while POMCP and DESPOT are relatively old algorithms, they still represent the canonical benchmarks to beat for online POMDP solvers especially for discrete models. See reference [9] for a recent survey of the field and various extensions of these main algorithms.
>
> RE Question 5. For problems such as Navigation1 where we need a long planning horizon and identification of the use of uncertainty reduction to solve the problem, general RL methods (including [2]) that do not represent uncertainty explicitly, when confronted with long horizons, will face difficulties. As for [1], it is actually very related to POMDP. Information state is an established concept that can be viewed as a more general representation to the concept of beliefs in POMDP[e.g.,  M. Hauscrecht. Value-Function Approximations for Partially Observable Markov Decision Processes. JAIR 2000]. In this sense, POMDP can be viewed as providing a more compact representation of uncertainty compared to  Information states. In fact, [1] proposes to frame PORL based on approximate POMDP planning. Therefore, we foresee that a method that improves POMDP solving or reformulates POMDP into something that can be solved with less computational resources could in turn be useful for PORL too.

---

### Official Review · Reviewer_NtUC · 2023-07-20

**Soundness:** 4 excellent
**Presentation:** 3 good
**Contribution:** 3 good
**Rating:** 7
**Confidence:** 3

**Summary:**

The paper is well written, clear and concise, with a well defined scope and goal.

This work extends prior work on Linearly Solvable MDPs (LS-MDPs) to the partially observable setting.  LS-MDPs are alternate decision processes whose control paradigm is shifted (w.r.t. standard MDPs) such that the system defines passive state dynamics (as in Markov chains or HMMs), and control is performed by applying smooth modifications to these passive dynamics that directly alter the state dynamics.  The theory of LS-MPDs already contains efficient solution methods for such decision processes.  The formulation of LS-MDPs can then be exploited to solve MDPs by embedding the MDP as an LS-MDP, and choosing the action from the action-set that most closely resembles the optimal control associated with the LS-MDP.

In this work, the authors extend LS-MDPs to the partially observable setting in a fairly straightforward way:  by applying the LS-MDPs framework to the belief-MDP associated with the POMDP at hand.  LS-MDPs operate in terms of state-to-state transitions and distributions U(s'|s), which in the POMDP case become belief-to-belief transitions U(b'|b), or, equivalently, U(a, o|b) (since the tuple (b,a,o) uniquely determines the next belief b').  Aside from this notational simplification, the resulting method seems to reflect the basic LS-MDP very straightforwardly.

Possibly the main difficulty remains that some computations remain complex in belief-space, as the authors also note, w.r.t W^*.  The authors therefore propose to employ sampling-based solvers to estimate W^*, resulting in RefSolver, their proposed planning algorithm for POMDPs.

- Minor notes:
-- Border cells are cut off from Figure 1;  remake graphics showing them in full.
-- Probably as a consequence of the above, the initial belief (in blue) is not visible in Figs 1a and 1b.


**Strengths:**

- theoretically sound
- shows significant improvements against POMCP baseline on reasonably stochastic and partially observable gridworld navigation tasks

**Weaknesses:**

- the evaluation is run only on 2d navigation tasks where the actions chosen by the optimal fully osbervable agent strongly correlate with the optimal actions chosen by a partially observable agent (see question below).  Given the role and influence that the optimal fully observable policy has on RefSolver, it's unclear how much the optimality of the passive policy influences the overall method.

**Questions:**

- both the POMCP baseline and the proposed RefSolver use the fully observable policy obtained by A* in one way or another.  POMCP uses it exclusively as a rollout policy, while RefSolver uses it in what seems a much more impactful manner by having it determine the passive dynamics of the reference-based POMDP formulation. WHat is the impact of this choice?  Would the algorithm work well enough even if a worse policy were chosen?  what about a random one?

**Limitations:**

- The final paragraph already identifies a few limitations of the proposed method concerning the computational methods employed.

- My question to the authors might also highlight another limitation of the resulting method, depending on the authors' answer.

---

> ### Author Rebuttal · Authors · 2023-08-08
>
> Thank you very much for the positive review and feedback.
>
> Just as a clarification, you mentioned in your review that the extension of LS-MDPs to partially observable domains is “very straightforward”.  We do note however that there are some interpretive challenges of formulating “belief-to-belief” stochastic transitions in the context of POMDPs as it requires relaxing the action space to provide for stochastic actions.  We believe that this insight is what makes the extension possible and useful and should be considered a contribution as well.
>
> RE “Remake graphics in Figure 1”: Note that each grid is 60x60 cells and so some cells look very small. The cells are there but are different grades of blue depending on the intensity of the belief. E.g. in Navigation1 the initial belief is spread out uniformly over 12 cells adjacent to S1 and S2 while in Navigation2 all the mass is concentrated on 2 cells adjacent to S1 and S2. We can make a note to explain this in the caption and increase the image sizes in the final version. There is no initial belief on Fig1b as this map is only used to construct the fully observed A* policy and is not used for planning under partial observations.
>
> RE Weaknesses: We recognise that the experimental results are preliminary at this early stage. We are currently working on how the algorithm could be scaled up to higher-dimensional navigation tasks as well as to motion planning tasks but the role of this paper was just to demonstrate that the approach is promising and to lay down the theoretical underpinnings. As mentioned in the general comments, we note again that even in this relatively simple environment, POMCP and DESPOT have difficulty managing the size of the search tree.  RefSolver on the other hand relies on analytical properties to selectively sample the tree.
>
> RE Questions: We do acknowledge that a judicial choice for the reference policy needs to be made as the problem formulation, by definition, rewards respecting the reference policy (see eq 10). However, a solution to a Ref-POMDP will also ignore the reference policy if greater rewards are available by deviating from it (also eq 10). As such, there is a level of robustness against the reference policy baked in to the formulation. We acknowledge the method should therefore be carefully applied to situations where it's apparent that a "good" reference policy exists (e.g. minimise energy) or can be justified (e.g. using an embedding as outlined in Sec 4). As noted in the summary, we envisage RefSolver to be suitable for problems with dynamically changing POMDP models where one needs to deform the policy being followed without making abrupt changes to the policy. We also hope that for more general problems we can substantially revise the algorithm by iteratively improving the reference policy.
>
> With respect to POMCP using the A* policy, this is an advantage over simply using a standard uniform rollout especially in a long-horizon problem. With the A* policy there is a good chance that the rollout will realise the large rewards toward the tail end of the time horizon, whereas there is almost no chance using a uniform rollout given the long horizon.
>
> As a general point, note that most complex problems in optimisation require a heuristic or initial policy that can be iteratively improved so we do not believe that the requirement of a reference policy is too restrictive.    Generally, as we suspect it is the case here, the proximity of the “guess” to the final solution also determines the speed of convergence.

---

> > ### Comment · Reviewer_NtUC · 2023-08-14
> >
> > Thank you for your response and clarifications;  my opinion of the submission was already positive to begin with and remains so having checked out the other reviews and rebuttals.  Best luck!

---

### Author Rebuttal · Authors · 2023-08-10

We would like to thank all the reviewers for their contributions and respond to some general points here.

First, we would like to emphasise again the key contributions of this paper.  The major contribution of the paper is that we propose a reformulation of POMDPs, such that the  optimisation for solving POMDPs no longer requires exhaustive enumeration of the action space (this enumeration is required by almost any POMDP planning method today). Instead, optimisation can be performed analytically as a consequence of a relaxation which, under suitable transformations, leads to a linearisation of the Bellman equation (Theorem 3.1) and allows the Bellman equation to be computed as an expectation, which in turn can be more efficiently computed using Monte Carlo methods relative to other state-of-the-art POMDP solvers.

We call this general reformulation of POMDPs a Reference-Based POMDP. The solution of a Reference-Based POMDPs needs to respect some given reference policy (e.g. heuristic, initial guess) while also searching for high rewards. A key idea in the formulation is to relax the space of optimisation such that policies are no longer mappings from beliefs to deterministic actions but rather stochastic actions.  This means that the Bellman value can be computed as an expectation under the reference stochastic policy, which leads naturally to Monte Carlo methods where the sample distribution is the reference policy itself.  As the reference policy is known a priori, the above insight facilitates algorithms where actions can be sampled, as opposed to enumerated, at each step.  This substantially reduces computation time as demonstrated in our preliminary algorithm. It is important to note that the mechanisms used in our algorithm is fundamentally different to that of the state-of-the art solvers.  Simulations resemble a depth-first search so that a sparse tree is constructed by sampling actions rather than a full enumeration as POMCP and DESPOT do.

In summary, our main contribution is a theoretical justification that decent (albeit suboptimal) reference policies can lead to fast algorithms for solving general POMDP problems substantially better than existing state-of-the-art solvers.  Note that the 2D problem we have presented, while simple, is not trivial even for state-of-the-art solvers like POMCP and DESPOT due to the long horizon and the enormous size of the search tree.  We alleviate this problem by sampling under the reference policy in our algorithm.

Finally, to place this appropriately in the literature, we are not aware of any papers that have extended Linearly Solvable MDPs (see Todorov [18-20]) to partially observable domains. Note that Linearly Solvable MDPs are closely related to Max Ent RL[2] where some recent theoretical developments have demonstrated robustness to the model (i.e. dynamics and rewards) in fully observable domains (see [2]) but again this has no parallel in partially-observable domains as of yet.

Re the Ethics Reviews, thank you for your insights and citations.  We do see, for instance, applications of this kind of framework in the context of artificial agents that are guided by a human to make responsible decisions while also judiciously transgressing guidance in the case of significant human error. Such systems may, at a future stage, be implemented in equipment that may be extremely difficult to manage without automated assistance and, of course, should be thoroughly tested and understood before final commissioning.  We agree it is appropriate to address your recommendations in the final version of the paper.

---

### Decision · Program_Chairs · 2023-09-21

**Decision:**

Accept (poster)

**Comment:**

This paper presents Reference-Based POMDPs, which is a reformulation of a standard POMDP where an agent tries to solve the POMDP while also trying to respect the reference policy. This new formulation also extends linearly solvable MDPs to the POMDP case. The formulation is described and solution methods, which no longer need to iterate over all actions, are introduced.

The setting and methods are novel and significantly different from standard POMDP ideas. There are also strong assumptions about having suitable reference policy and belief dynamics, but the method seems promising for improved solutions in a large class of realistic POMDPs.

Because the method is very different from standard POMDP literature, more details and discussion of related work should be included. Currently, it is difficult to follow some of the details and assumptions in the method. The author response and discussion were helpful in this regard but these clarifications should be included in the paper.

Furthermore, there are some concerns about scalability and generality of the method. More comprehensive results with larger and more complex domains and different quality reference policies would be helpful but, at the least, discussion about these topics should be included.